

# Rhizocompartmental microbiomes of arrow bamboo (*Fargesia nitida*) and their relation to soil properties in Subalpine Coniferous Forests

Nan Nan Zhang[1,2], Xiao Xia Chen[1,2,3], Jin Liang[1,2], Chunzhang Zhao[4], Jun Xiang[1], Lin Luo[1,2,3], En Tao Wang[5] and Fusun Shi[1,2]

[1] Chengdu Institute of Biology, Chinese Academy of Sciences, Chengdu, China
[2] CAS Key Laboratory of Mountain Ecological Restoration and Bioresource Utilization & Ecological Restoration and Biodiversity Conservation Key Laboratory of Sichuan Province, Chengdu Institute of Biology, Chinese Academy of Sciences, Chengdu, China
[3] University of Chinese Academy of Sciences, Beijing, China
[4] Chengdu University of Technology, Chengdu, China
[5] Escuela Nacional de Ciencias Biológicas, Instituto Politécnico Nacional, Ciudad de México, México

Corresponding authors
Chunzhang Zhao,
zhaochzh04@126.com
Fusun Shi, 3651230347@qq.com

## ABSTRACT

Arrow bamboo (*Fargesia nitida*) is a pioneer plant in secondary forest succession in the Sichuan Province mountains. To comprehensively investigate the microbial communities and their functional variations in different rhizocompartments (root endosphere, rhizosphere, and root zone) of arrow bamboo (*Fargesia nitida*), a high-throughput metagenomic study was conducted in the present study. The results showed that the abundances of the dominant bacterial phyla Proteobacteria and Actinobacteria in the bamboo root endosphere were significantly lower than those in the rhizosphere and root zones. In contrast, the dominant fungal phyla, Ascomycota and Basidiomycota, showed the opposite tendency. Lower microbial diversity, different taxonomic composition and functional profiles, and a greater abundance of genes involved in nitrogen fixation (*nifB*), cellulose degradation (beta-glucosidase), and cellobiose transport (cellulose 1, 4-beta-cellobiosidase) were found in the bamboo root endosphere than in the other rhizocompartments. Greater soil total carbon, total nitrogen, $NH_4^+$-N, microbial biomass carbon, and greater activities of invertase and urease were found in the bamboo root zone than in the adjacent soil (spruce root zone). In contrast, the soil microbial community and functional profiles were similar. At the phylum level, invertase was significantly related to 31 microbial taxa, and the effect of $NH_4^+$-N on the microbial community composition was greater than that of $NO_3^-$-N. The soil physicochemical properties and enzyme activities were significantly correlated with microbial function. These results indicate that the root endosphere microbiomes of arrow bamboo were strongly selected by the host plant, which caused changes in the soil nutrient properties in the subalpine coniferous forest.

## INTRODUCTION

Plant-associated microbiomes have an obvious influence on the nutrition, growth, and health of plants, the structure of plant communities, and the resistance and adaptation of plants to environmental changes (*Ke, Miki & Ding, 2015*). Plants can also regulate microbial activity and community composition in the rhizosphere by secreting root exudates or bioactive molecules (*Dhungana, Kantar & Nguyen, 2023*; *Hu et al., 2018*). Another type of root-associated microbiome is endophytes, which colonize the root endosphere to obtain a stable habitat and nutrient supply and exert more lasting effects than rhizospheric microbes (*Hong et al., 2015*). Endophytes have multiple functions in plant physiology, such as antibiotic activity, phytohormone production, and immunity enhancement, which can promote plant growth and mitigate biotic and abiotic stresses in plants (*Dini-Andreote, 2020*; *Jasim et al., 2014*).

Previously, a hierarchical decrease in microbial diversity in rhizocompartments toward the roots has been described, suggesting that plants filter and recruit their microbiome subsets (*Bulgarelli et al., 2012*). A two-step model for microbiome selection by plants has been suggested (*Barajas et al., 2020*; *Bulgarelli et al., 2013*), which includes (1) chemotaxis of microbes from bulk soil to the rhizosphere caused by root exudates, which is the first step in selecting rhizosphere microbes from bulk soil; and (2) selective entry of microbes to the root endosphere regulated by plant genetic factors, such as host plant defense strategies, root structure, and root exudation (*Bulgarelli et al., 2012*), which further select endosphere microbes from the rhizosphere community. According to this model, the composition of microbial communities in the root endosphere, rhizosphere, and bulk soil could be substantially different from each other, and the filtration effects followed the order of endosphere >rhizosphere >root zone (*Xiao et al., 2017*).

Arrow bamboo in the genus *Fargesia*, a semi-woody plant, is a rhizomatous perennial species that often dominates the plant community in the understory of some montane species in East Asia and South America (*Liu et al., 2014*; *Saitoh, Seiwa & Nishiwaki, 2002*). The dense leaves of arrow bamboo form a barrier for penetrating sunlight to the lower layers, which could inhibit the survival and regeneration of tree seedlings, saplings, and mature trees in many forests (*Takahashi et al., 2003*; *Wang et al., 2007*). Furthermore, *Fargesia* bamboo is the main food source of the endangered giant panda (*Wang, Tao & Zhong, 2009*). In southwest China, arrow bamboos are widely distributed understory plants in the subalpine spruce forest, a primary representative subalpine coniferous forest after the clearing of old-age dark coniferous forests since the 1950s (*Ma et al., 2007b*; *Zhang, Liu & Gu, 2011*). Arrow bamboo is a pioneer plant in various secondary forest communities in natural succession after local coniferous forests are destroyed (*Ma et al., 2007a*; *Xu et al., 2016*). In addition to its negative effects on the diversity of understory species and forest regeneration caused by its high density (*Wang, Shi & Tao, 2012*), arrow bamboo might also change soil physicochemical traits, as reported for other bamboo species (*Kaushal et al., 2020*; *Umemura & Takenaka, 2015*), which might affect the soil microbial communities and subsequent plants.

Reportedly, the bacterial microbiomes of moso bamboo (*Phyllostachys edulis*) are structurally variable in both the rhizosphere and the endosphere in Guangxi Province, China (*Yuan et al., 2021*). However, the filtration (selection) of arrow bamboo rhizocompartments and the interactions of the bamboo microbiome with soil characteristics in subalpine coniferous forests have not been described. Therefore, we conducted the present study to (i) investigate the diversity and function of microbiomes in the root zone, rhizosphere, and endosphere of arrow bamboo in the subalpine spruce forest to reveal the filtration function of rhizocompartments and (ii) analyze the linkage between soil properties and microbial communities.

## MATERIALS AND METHODS

### Study site

The experiment was conducted at the Maoxian Mountain Ecosystem Research Station of the Chinese Academy of Sciences (103°54′ E, 31°42′ N) on the eastern edge of the Tibetan Plateau of China. Located in a montane temperate climate region, the sampling site presented a mean annual temperature of 9.3 °C, annual precipitation of 850 mm, mean annual evaporation of 796 mm, and Calcic Luvisol soil (*Food and Agriculture Organization of the United Nations (FAO), 2006*). Local spruce (*Picea asperata* Mast.) plantations were established in the 1980s as a typical cultivated tree species in the eastern Tibetan Plateau subalpine region. Arrow bamboo, *Fargesia nitida* (Mitford) Keng f. ex T.P. Yi, is the dominant species scattered throughout the shrub layer of the spruce alpine forest. The other epidemic understory plants in this region are *Asparagus filicinus* D. Don, *Phlomis umbrosa* Turcz., *Thladiantha davidii* Franch., and *Sinosenecio oldhamianus* (Maxim.) B. Nord., *Plantago major* Linn., *Carpesium divaricatum* Sieb. & Zucc., *Thalictrum uncinulatum* Franch., *Paraprenanthes melanantha* (Franch) Ze H. Wang, and *Rosa sericea* Lindl. (*Hu et al., 2016*). The samples were collected from the Maoxian Mountain Ecosystem Research Station of the Chinese Academy of Sciences with field permit number 2019016. In this study, the treatments (samples) were (1) arrow bamboo root endosphere (BE), (2) arrow bamboo rhizosphere (BR), (3) arrow bamboo root zone (BZ), and (4) spruce root zone (SZ), which also serves as a control of the soil. Soils and roots were sampled at the end of the growing season on October 11, 2019. Nine replicate plots (400 m² each) were set-up in the alpine forest (mixed-coniferous forest).

From each plot, the root systems of five randomly selected *F. nitida* plants were removed. Because most of the roots were distributed in the 0–20-cm soil layer, the root zone soils were also sampled from this layer. After removing the soil particles loosely attached to the roots by vigorous shaking, the soil that tightly adhered to the root surface was brushed off as rhizosphere soil. At the same time, a composite soil sample for each plot was taken as the root zone soil (<20 cm from the collar of the sampled bamboo). Correspondingly, nine spruce root zone soils were collected from the same plots (approximately 20 m from the sampled bamboo to weaken the rhizosphere effect of the bamboo plant). Surface soils (0–20 cm in depth) were sampled from a 20 cm × 20 cm area with a distance of 0.5 m from the spruce tree. All root and soil samples were transported to the laboratory, and parts
were stored at 4 °C and −20 °C after passing through a 2-mm sieve for further analysis. Another portion of each soil sample was air-dried for physicochemical analysis.

## Analysis of soil physicochemical properties and enzyme activity

The soil samples were passed through sieves with a width of 0.15 mm. Total carbon (TC) and total nitrogen (TN) were determined using an elemental analyzer (Vario MACRO cube CN, Elementar Analysensysteme, Germany) (*Wang et al., 2016*). Soil $NH_4^+$-N and $NO_3^-$-N were extracted using 1 M KCl and measured by SEAL AA3 continuous flow analysis (SealAnalytical, Norderstedt, Germany). Available phosphorus (AP) was evaluated using the Olsen method with inductively coupled plasma combined with optical emission spectroscopy (*Olsen & Cole, 1954*). The standard method of Mc *Lean & Watson (1985)* was used to determine available potassium (AK) content. Soil electrical conductivity (EC) and pH were measured in 1:5 and 1:2.5 (w/w) soil: water suspensions, respectively. Soil microbial biomass carbon (MBC) and nitrogen (MBN) were measured using the chloroform fumigation extraction method (*Vance, Brookes & Jenkinson, 1987*), using Kec and Ken factors of 0.45 and 0.54 for extractions of carbon and nitrogen, respectively.

Invertase activity was measured according to the method of *Xiao et al. (2017)*, based on the absorbance measurement at 508 nm of glucose released, which was reacted with 3,5-dini-trosalicylic acid. To assay the urease activity, 5 g of air-dried soil was incubated with 10 mL of 10% (w/v) urea solution, one mL of toluene, and 20 mL of citrate buffer (pH 6.7) at 37 °C for 24 h. Urease activity was expressed as the ammonium released in mg $g^{-1}$ soil at 24 h (*Xiao et al., 2017*). Protease activity was measured using the method described by Ladd and Butler (1972). Protease activity was expressed as the amount of ammonium released ($\mu g\ g^{-1}$ soil $h^{-1}$). Cellulase activity was expressed as milligrams of glucose equivalents per gram of soil per hour (*Guan, Zhang & Zhang, 1986*).

## Sample preparation and metagenomic DNA extraction

Surface sterilization and decontamination of surface DNA were performed on root samples according to a previously described protocol (*Woźniak et al., 2019*; *Yang et al., 2001*). The endophytic microbiome DNA (0.5 g of each root sample) was extracted using an EZNA™ plant DNA kit (Omega Bio-Tek, Norcross, GA, USA). Metagenomic DNA (0.5 g) from each rhizosphere and root zone soil sample was extracted using the EZNA™ Omega Mag-bind soil DNA kit in triplicate (Omega Bio-Tek, Norcross, GA, USA), according to the manufacturer's instructions. The quantity and quality of the DNA extracts were evaluated using a NanoDrop spectrophotometer (ND-1000; Thermo Fisher Scientific, Waltham, MA, USA) and 1% (w/v) agarose gel electrophoresis, respectively.

## Sequencing and bioinformatic analyses

Initially, 36 metagenomic DNA libraries (four treatments, nine duplicates) for shotgun sequencing were prepared using Illumina TruSeq Nano library preparation kits with insert sizes of 400 bp. Each library was sequenced on an Illumina NovaSeq platform (Illumina, Foster City, CA, USA) using the PE150 strategy at Personal Biotechnology Co., Ltd. (Shanghai, China). Raw sequencing data were processed to obtain quality-filtered reads as follows: adapter sequences were trimmed using Cutadapt (v1.2.1) (*Martin, 2011*);
(2) sequences with low quality (quality value <20 or contain N bases) and short reads (<50 bp) were removed; (3) Chloroplast DNA and mitochondrial DNA sequences were removed using Burrows–Wheeler Aligner (BWA) (*Li & Durbin, 2009*). The quality-filtered reads remained for *de novo* assembly using IDBA-UD (*Peng et al., 2012*) with the following parameter settings: -mink 40, -maxk 120, -min_contig 200. Coding genes were predicted using MetaGeneMark on scaffolds of >300 bp (*Zhu, Lomsadze & Borodovsky, 2010*). A non-redundant gene catalog was constructed using CD-HIT, with a sequence identity cutoff of 0.95 and a minimum coverage cutoff of 0.9 for the shorter sequences. Gene abundance in each sample was calculated using SOAP alignment (identity = 0.95). Species rarefaction curves were generated to assess the sequencing depth using QIIME 1.9.1. Alpha diversity indices (Shannon, ACE, and Chao1) were compared using QIIME 1.9.1. Microbial profiles of the taxonomy of the non-redundant genes were classified by aligning the obtained sequences against the NCBI NT database using BLASTN (*e*-value <0.001). DIAMOND (v0.9) was used to obtain functional profiles of the non-redundant genes by aligning them with the KEGG database and the Carbohydrate-Active EnZymes database (CAZy) (*Buchfink, Xie & Huson, 2015*). All acquired raw sequences in this study have been deposited in the Sequence Read Archive of the Chinese National Microbiology Data Center with the accession number NMDC40026007–NMDC40026042.

## Statistical analyses

All statistical analyses were performed using R version 4.0.2 (*R Core Team, 2020*). Differences in the composition and diversity of microbiomes among the rhizocompartments of bamboo were statistically compared by ANOVA using Duncan's multiple range test. The differences between the root zone soils of bamboo and spruce were evaluated using the function "t.test" in the *stats* package. Taxonomic and functional structure was visualized using principal coordinate analysis (PCoA) on Bray–Curtis distances. PERMANOVA was conducted to test the differences in beta diversity among rhizocompartments using the "adonis" function in "vegan." The correlation between the microbial community (at the phylum level) and soil physicochemical properties was tested using the Pearson coefficient. The effects of soil physicochemical properties on the soil microbial community were tested *via* redundancy analysis (RDA) using the vegan package in R. The correlations among the soil properties, genus-level taxonomic composition of microbiomes, and gene family distances (CAZy and KEGG) in the metagenome were determined using the Mantel test in the R vegan package (*Dixon, 2003*).

## RESULTS

### Comparison of soil physicochemical properties

The results in Fig. 1 show that TN, TC, $NH_4^+$-N, MBC content, and the activities of invertase and urease were significantly higher in the bamboo root zone than in the spruce root zone soils. No obvious differences in the MBN and NO3–N contents, pH and EC values, or protease and cellulase activities were observed between these two soil samples.
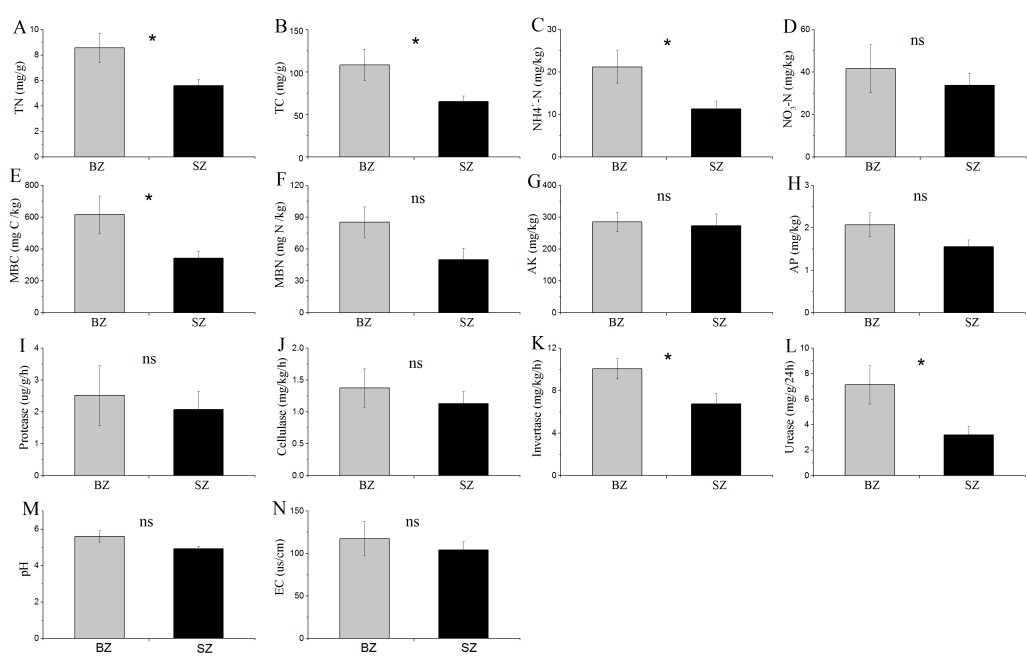

**Figure 1** **Comparative analysis of (bio-)chemical factors root zone soils.** BZ, arrow bamboo root zone; and SZ, spruce root zone; TN, total nitrogen; TC, total carbon; $NH_4^+$-N, ammonium nitrogen; $NO_3^-$-N, nitrate nitrogen; MBC, microbial biomass carbon; MBN, microbial biomass nitrogen; AK, available K; AP, available P; EC, electrical conductivity; ns, not significant; *, $P < 0.05$.

## Microbial community composition

A total of 3,068,211,656 bp of DNA sequences were obtained after quality control, and a summary of the sequencing data is presented in Table S1. Rarefaction curves (Fig. S1) indicated that sufficient sequencing depth was achieved in this study. There were significant differences in the taxon numbers at the domain level of the microbiomes among the root endosphere and soil samples (Table S2). Bacterial taxa accounted for 74.45% of the microbes in root endosphere of bamboo, which was significantly lower than that in the rhizosphere and root zone, whereas eukarya showed the opposite pattern. Archaea accounted for 0.07% of the microbes in root endosphere, being significant lower in the root endosphere than that in the rhizosphere and root zone of bamboo. No significant differences were detected in archaea, bacteria, or eukarya between the root zone soils of bamboo and spruce. The root endosphere and soil microbiomes were dominated by Proteobacteria (59.81%–71.84%), Actinobacteria (6.45%–13.01%), Acidobacteria (1.90%–5.03%), Ascomycota (0.16%–3.12%), and Planctomycetes (1.20%–1.74%), although the relative abundances varied among the samples (Fig. 2 and Table S3). The abundance of dominant bacterial phyla (such as Proteobacteria and Actinobacteria) in the root endosphere was significantly lower than that in the rhizosphere and root zones. Abundances of the nine most dominant fungal phyla, Ascomycota (comprising 56.65%–59.71% of fungal sequences) and Basidiomycota (21.98%–32.46% of fungal sequences) were significantly greater in the endosphere of bamboo than in the other rhizocompartments (Fig. 2). Representatives of archaea, such

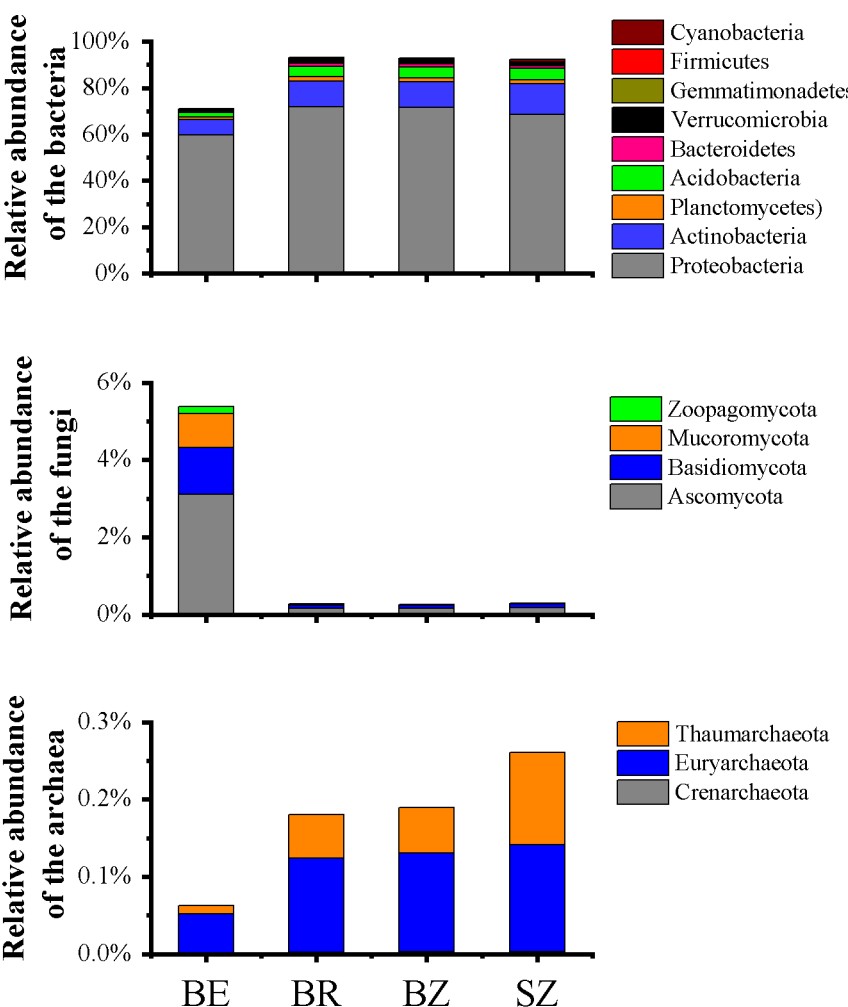

**Figure 2** **Relative abundances of the bacterial (A), fungal (B) and archaeal (C) phylotypes in the metagenomes of microbiome in each sample, and their major variations in root endosphere, rhizosphere and root zone soils.** The other Eucaryota and viruses occupied the remaining parts of the relative abundances of the sequencing data (Table S2). Relative abundances are calculated as the proportion of individual taxa to the total number of corresponding phylotypes reads. BE, arrow bamboo root endosphere; BR, arrow bamboo rhizosphere; BZ, arrow bamboo root zone; and SZ, spruce root zone.

as Euryarchaeota, Thaumarchaeota, and Crenarchaeota, were significantly less abundant in the root endosphere than in the other rhizocompartments. At the phylum level, the abundance of bacteria in Aquificae, Thermotogae, Thermodesulfobacteria, Deferribacteres, and fungi in Mucoromycota significantly differed between the root zone soils of bamboo and spruce. As the dominant genera among the samples, *Pseudomonas* and *Burkholderia* exhibited significantly higher abundances in the root endosphere than in the other rhizocompartments. Similar results were observed in *Klebsiella*, *Massilia,* and *Collimonas*, whereas *Bradyrhizobium*, *Rhizobium,* and *Streptomyces* values were reversed (Table 1). Higher abundances of *Pseudomonas* and *Massilia* were observed in the bamboo root zone than in the spruce root zone.

**Table 1** Comparative analysis of relative abundance of the domain (top 15) genera in the roots and soils.

| Genus | Domain | BE (%) | BR (%) | BZ (%) | SZ (%) |
|---|---|---|---|---|---|
| *Pseudomonas* | Bacteria | 7.87a | 1.94b | 1.82b | 1.57[*] |
| *Bradyrhizobium* | Bacteria | 12.63b | 15.63a | 15.44a | 15.20 |
| *Burkholderia* | Bacteria | 3.46a | 2.62a | 2.57a | 2.51 |
| *Mesorhizobium* | Bacteria | 0.81a | 1.07a | 1.02a | 1.03 |
| *Variovorax* | Bacteria | 0.57a | 0.71a | 0.71a | 0.56 |
| *Rhodoplanes* | Bacteria | 0.90b | 2.19a | 2.14a | 1.88 |
| *Rhizobium* | Bacteria | 0.56b | 0.74a | 0.74a | 0.75 |
| *Streptomyces* | Bacteria | 1.14b | 1.69a | 1.67a | 2.00 |
| *Paraburkholderia* | Bacteria | 1.05a | 0.85a | 0.78a | 0.71 |
| *Klebsiella* | Bacteria | 0.48a | 0.06b | 0.08b | 0.07 |
| *Massilia* | Bacteria | 0.58a | 0.27b | 0.26b | 0.21[*] |
| *Sphingomonas* | Bacteria | 0.49b | 0.73a | 0.73a | 0.71 |
| *Collimonas* | Bacteria | 0.83a | 0.25b | 0.23b | 0.16 |
| *Rhodopseudomonas* | Bacteria | 0.56b | 1.02a | 1.01a | 0.97 |
| *Mortierella* | Eukaryota | 0.61a | 0.33b | 0.2b | 0.18 |

Notes.

Different lowercase letters represent significant differences among bamboo samples using Duncan's multiple range test. T test was used for comparing root zone soil of bamboo and spruce.

[*]$p < .05$.

BE, bamboo root endosphere; BR, bamboo rhizosphere; BZ, bamboo root zone; SZ, spruce root zone.

As estimated by the Shannon, ACE, and Chao1 indices, microbial diversity in the root endosphere was significantly lower than that in the other bamboo rhizocompartments (Fig. 3). ACE richness was significantly higher in the bamboo root zone than in the spruce root zone. The composition of the microbial communities in the bamboo root endosphere differed from that in the other rhizocompartments. In contrast, the root zone soil compositions of bamboo and spruce were similar, as shown in the PCoA plot and PERMANOVA analysis (Fig. 4A and Table S4). Compared with the rhizosphere and root zone of bamboo, 243 and 242 significantly enriched genera, and 900 and 909 significantly depleted genera were detected in the bamboo root endosphere (Fig. S2). Compared with the microbiomes in the spruce root zone, 247, five, and three significantly enriched genera and 947, 55, and 59 significantly depleted genera were detected in the bamboo endosphere, rhizosphere, and root zone, respectively (Fig. S3). Furthermore, 77 genera were specific to spruce soils, and 116 genera were specific to bamboo soil, but all of them were minor groups with relative abundances ≤1 ppm (J. Xiang, 2021, pers. obs.).

## Functional gene profiles of microbial communities

Using the KEGG and CAZy databases, 12,465 KO genes (Fig. 5) among the metagenomic reads and 332 CAZy (Fig. 6) functional genes related to carbon and nitrogen cycles were screened, respectively, in all the samples. Functional changes between root and soil microbiomes were investigated *via* PCoA and PERMANOVA analysis (Figs. 4B and 4C, Table S4), which showed a significant difference between the endosphere and the other bamboo rhizocompartments but was similar between the bamboo root zone and spruce

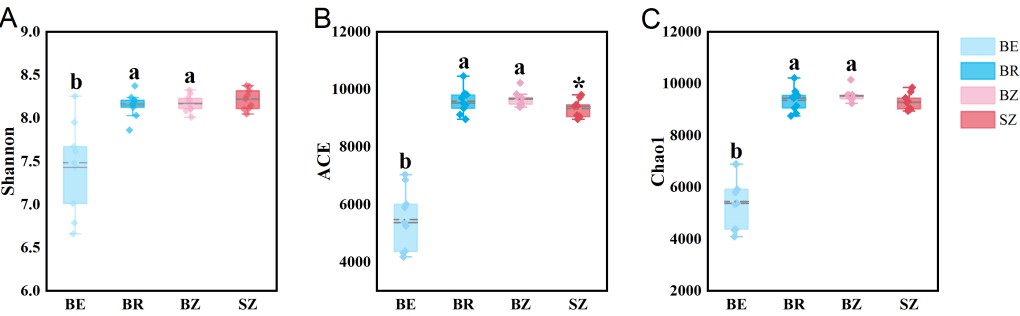

**Figure 3** **The microbial community diversity in the roots and soils.** Different lowercase letters represent significant differences among bamboo samples using Duncan's multiple range test. $T$ test was used for comparing root zone soil of bamboo and spruce * $p < .05$. BE, bamboo root endosphere; BR, bamboo rhizosphere; BZ, bamboo root zone; SZ, spruce root zone.

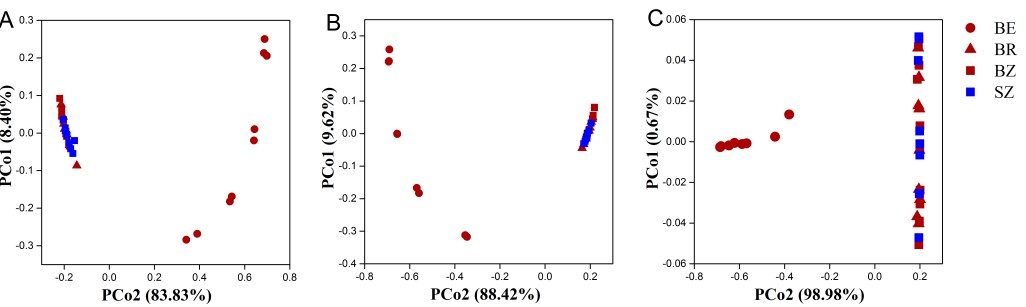

**Figure 4** **The taxonomic (A: the genus level) and functional (B: KEGG pathway level and C: CAZy level) structure of microbiomes in the roots and soils.** BE, arrow bamboo root endosphere; BR, arrow bamboo rhizosphere; BZ, arrow bamboo root zone; and SZ, spruce root zone.

root zone (KEGG and CAZy databases). The relative abundances of genes involved in the carbon and nitrogen cycles in the root endosphere were different from those in the other bamboo rhizocompartments (Fig. 5). For example, the abundance of genes encoding the nitrogen fixation protein NifB, beta-glucosidase, and cellulose 1, 4-beta-cellobiosidase in the root endosphere was 10-, 200-and 39-fold higher, respectively, than those in the other rhizocompartments (Table S5). In addition, the abundance of some genes involved in multiple sugar transporters in the root endosphere was significantly different (greater or less depending on the proteins) from those in the rhizosphere and root zone of bamboo. The abundance of nitrate reductase changed significantly between the root zone soils of bamboo and spruce.

By annotating metagenomic sequences to the CAZy database, the most abundant enzyme class in the soil samples was the glycosyl transferases (GTs) (relative abundance, 37.15%–48.55%), followed by glycoside hydrolases (GHs) (relative abundance, 31.31–33.80%; Table S6). The relative abundances of GH, GT, and polysaccharide lyase (PL) classes were greater in the root endosphere than in the other rhizocompartments of bamboo. In contrast, those of auxiliary activities (AA), carbohydrate-binding modules (CBM), and

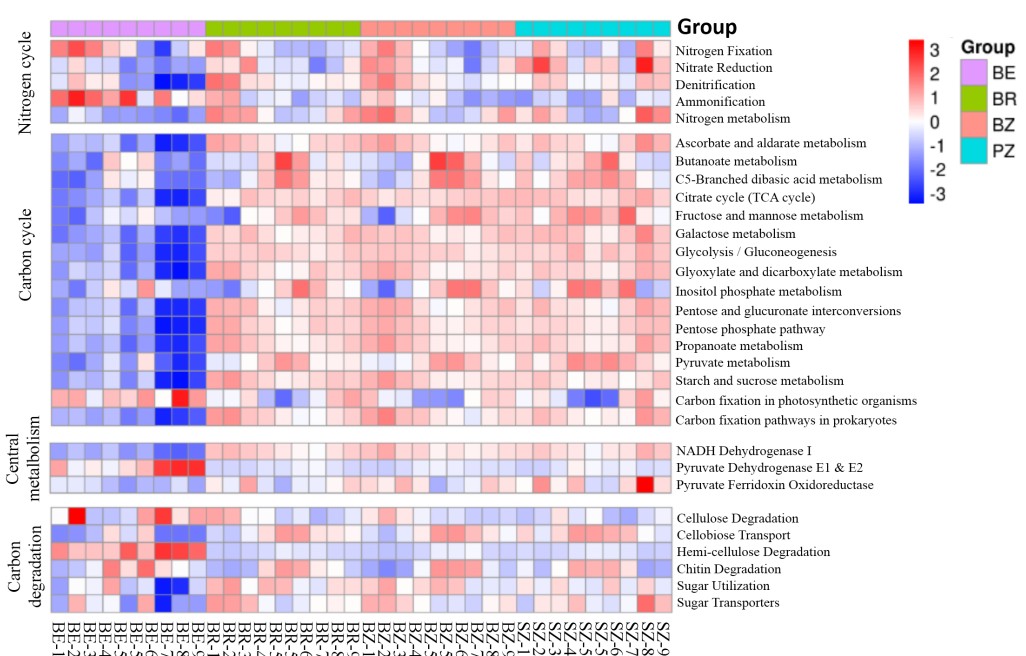

**Figure 5** **Heatmaps indicating functional genes abundances of the C and N cycles in the KEGG database that differ between the roots and soils.** BE, arrow bamboo root endosphere; BR, arrow bamboo rhizosphere, BZ, arrow bamboo root zone; and SZ, spruce root zone.

carbohydrate esterases (CE) classes were reversed ($P < 0.05$). The 50 dominant CAZy families in the root endosphere were significantly different from those in the rhizosphere and root zone of bamboo (Fig. 6). In contrast, most of these families did not vary between the root zone soils of bamboo and spruce (Table S7).

## Relationship between soil properties and microbiomes

Mantel tests between soil properties and microbial community distance matrices demonstrated that the soil properties were not correlated with shifts in the microbial community but were significantly correlated with the functional composition of microbiomes in the bamboo and spruce root zone soils (Table 2). The Spearman correlation coefficient revealed significant correlations between the soil properties and significantly varied microbial taxa (Fig. 7). In this study, 651 sequences belonging to 42 genera (including *Pseudacidobacterium* and *Flavihumibacter* as the dominant genera) were correlated with invertase (K01193), and 5,050 sequences belonging to more than 100 genera (including *Pseudomonas*, *Pseudonocardia*, *Streptomyces*, *Rhodoplanes*, and *Microvirga*) were correlated with urease (K01427, K01428, K01429, K01430, and K14048). Cellulases and proteases also correspond to many KOs belonging to over 100 genera. Invertase was significantly related to 31 microbial taxa (at the phylum level), and the effect of $NH_4^+$-N on microbial taxa was greater than that of $NO_3^-$-N. Soil physicochemical properties and enzyme activities were negatively associated with Crenarchaeota (archaea) and Elusimicrobia (bacteria). In the CAZy database, pH, TN, TC, $NH_4^+$-N, MBC, urease, and invertase significantly influenced

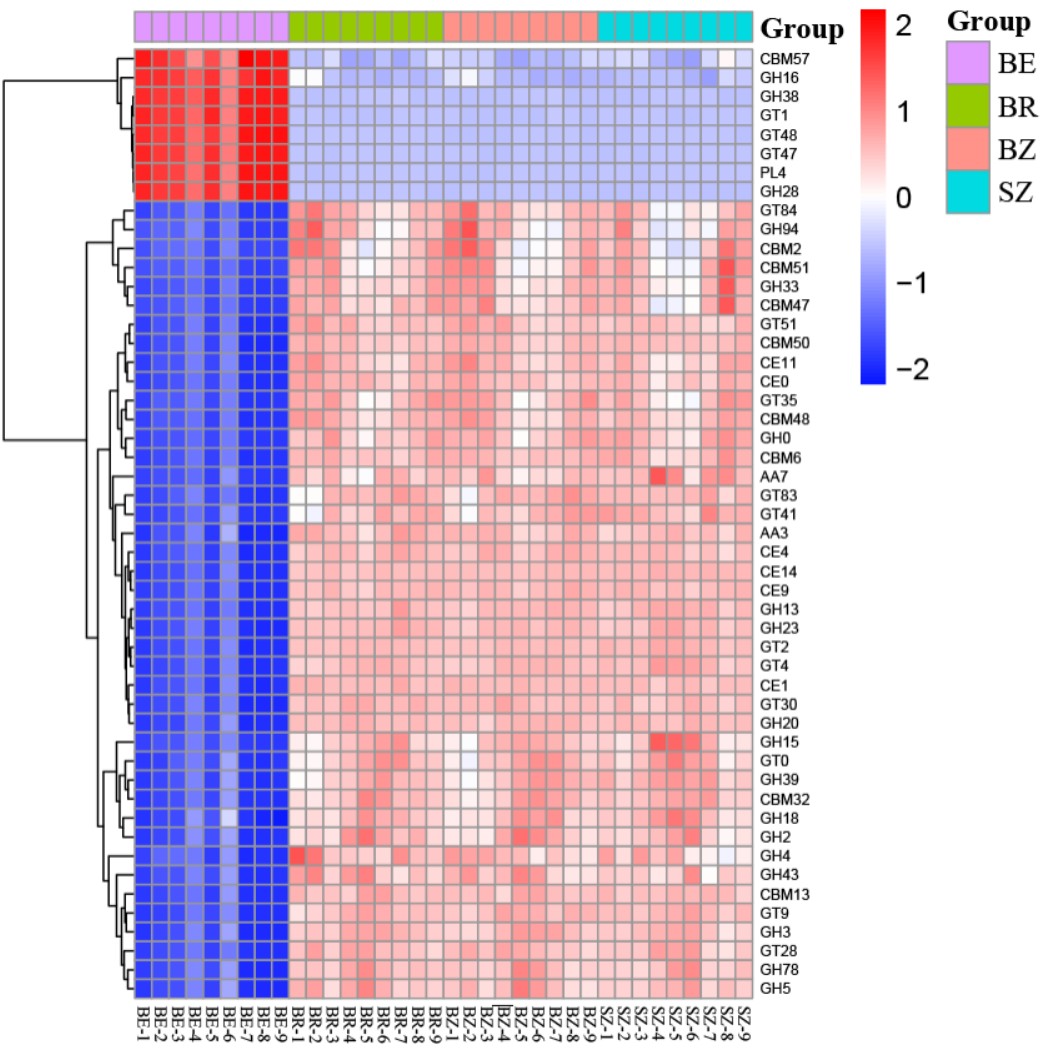

**Figure 6** The heatmaps show the distribution of gene classes (A) and the 50 most abundant gene families (B) in the CAZy database between the roots and soils. BE, arrow bamboo root endosphere; BR, arrow bamboo rhizosphere; BZ, arrow bamboo root zone; and SZ, spruce root zone.

CAZyme families (Table S8). The RDA results showed that soil pH and invertase were the main contributors to microbial taxa and KEGG compositions, and invertase and TN were the main contributors to CAZyme families (Fig. S4).

## DISCUSSION

### Filtration of rhizocompartments for microbiomes

The results obtained in the present study (Figs. 2–4) and in previous studies (*Woźniak et al., 2019*; *Xiao et al., 2017*; *Zhou et al., 2020*) demonstrated that root filtration significantly reduced biodiversity, represented by Shannon, ACE, and Chao1 indices (Fig. 3), as well as shifted the composition (Figs. 2 and 4) and functional patterns (Figs. 5 and 6) of microbiomes in the root endosphere. Furthermore, hierarchical selection in the other

**Table 2** Correlation among the soil properties, microbial communities, and the gene families (KEGG level 3 and CAZy level 2) estimated from the metagenomes of bamboo and spruce root zoon soils.

| | Soil properties[a] | Microbial communities[b] | KEGG genes |
|---|---|---|---|
| Microbial communities | 0.2199 | | |
| KEGG genes | 0.1855[*] | 0.6935[**] | |
| CAZy genes | 0.1713[*] | 0.6173[**] | 0.9623[**] |

**Notes.**
[a] Soil properties include TC, TN, $NH_4^+$-N, $NO_3^-$-N, MBC, MBN, AK, AP, pH, EC, protease, cellulose, invertase and urease.
[b] Microbial communities refer to the Bray Curtis distances of microbial communities (genus composition).
[*] indicates $P < 0.05$.
[**] indicates $P < 0.01$.

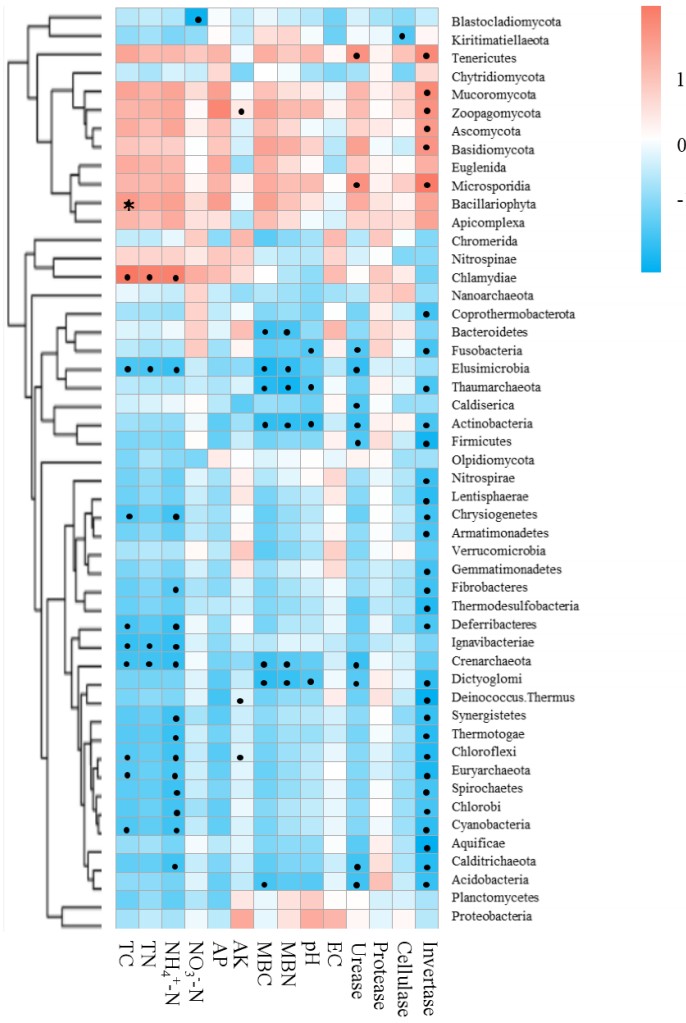

**Figure 7** Correlations between environmental factors and significantly changed microbial taxa of root zone soils using the Spearman correlation coefficient (red, positive; blue, negative). * $P < 0.05$.

rhizocompartments (bamboo rhizosphere, bamboo root zone, and spruce root zone) was not reflected in the biodiversity indices, community composition, and functions of the microbiomes (Figs. 3–5). Previously, the bacteriome structure in the endosphere, which is different from that in the rhizosphere, was also reported for microbiomes associated with Moso bamboo (*Yuan et al., 2021*). It was estimated that the difference was related to the rhizosphere physicochemical properties, the effects of roots on nutrient supply, and the environmental variation from the soil to the endosphere (*Gottel et al., 2011*). The lower abundance of the predominant bacterial phyla Proteobacteria and Acidobacteria in the root endosphere of arrow bamboo than in the rhizosphere might also result from root filtration. Although the abundance of dominant bacterial phyla decreased in the endosphere, the abundance of the main fungal phyla increased, suggesting the benefit of fungal endophytic colonization (*Yan et al., 2019*) by arrow bamboo roots from the surrounding soil fungal communities (*Bulgarelli et al., 2012*). Another possible reason for the increased abundance of fungi in the root endosphere could be related to mycorrhizal symbiosis, as *F. nitida* forms mycorrhiza (*Luo et al., 2022*), possibly with enrichment of Ascomycota, Basidiomycota, and Mucoromycota in the root endosphere (*Van der Heijden et al., 2015*; *Bonfante & Venice, 2020*).

In the present study, *Pseudomonas* was the predominant genus in all samples, but its abundance in the arrow bamboo root endosphere was significantly greater than that in the rhizosphere and root zone soils (Table 1). This finding was consistent with our isolation results (*Zhang et al., 2023*), indicating that plants of arrow bamboo prefer these bacteria as their endophytes. The endophytic *Pseudomonas* isolated from roots of bamboo (species name not reported) grown in Aceh Province of Indonesia presented good potential as a biological control agent against nematodes of *Meloidogyne* spp. (*Maulidia & Sriwati, 2020*). Endophytic *Pseudomonas* members isolated from bamboo (*Bambusa blumeana*) in Guangdong Province of China are renowned for their plant growth-promoting and nitrogen-fixing properties (*Wei et al., 2010*). In addition, many endophytes of arrow bamboo roots contain yolk-degrading bacteria and produce IAA (*Zhang et al., 2023*). Therefore, endophytic *Pseudomonas* may directly or indirectly stimulate bamboo growth and pathogen resistance (*Guzmán-Guzmán & Santoyo, 2022*).

The dominant bacterial genera in arrow bamboo, *Pseudomonas, Burkholderia,* and *Klebsiella*, are frequent endophytes of bamboo species (*Liu et al., 2017*; *Singh et al., 2021*). Bamboo endophytes may also colonize the guts of arrow bamboo eaters, such as giant pandas and insects (*Yao et al., 2021*). Being colonized in guts, these endophytes could help animals digest plant fibers or antimetabolites and synthesize metabolites with bioactivities (*Martínez-Romero et al., 2021*). It has been reported that endophytic bacteria in *Pseudomonas, Burkholderias,* and *Enterobacter* can digest some antimetabolites in their host plants (*Shanmuganandam et al., 2019*). In the gut metagenomes of the panda, an arrow bamboo eater, most genes related to the degradation of plant secondary metabolites are associated with *Pseudomonas* (*Zhu et al., 2018*). Identifying genes related to microbe-plant interactions and degradation of cellulose in *Klebsiella* strains isolated from panda faeces also indicated their endophytic origin and possible contribution to the digestion of cellulose in the panda gut (*Lu et al., 2015*). These findings enlarged the impact of plant endophytes on

herbivore digestion and implied that endophytes might be a determinant of the feeding habits of some animals.

This is the first report on the primary understanding of microbial function in the root and root-associated soils of arrow bamboo by analyzing the abundance of gene families and the functional classification schemes of the KEGG and CAZy databases. Endophytes commonly contain enzymes that can degrade plant cell wall composition, such as endoglucanase, cellulase (*Sharma et al., 2020*), and chitin degradation enzymes (*Proença et al., 2018*). Overall, the functional gene contents of soil microbial communities were significantly different between the endosphere and the rhizosphere (Figs. 5–6), indicating that the filtration effects for microbes by the rhizocompartments also change the metabolism, biosynthesis, and degradation processes in the carbon and nitrogen cycles of arrow bamboo.

## Soil and microbial properties of arrow bamboo and spruce root zone

Arrow bamboo is the pioneer and companion species of secondary forest communities in the subalpine spruce plantations of southwest China. Different plant species in aboveground systems can affect soil properties and root-associated microbes, mainly through root activity and litter input (*Diao et al., 2020*; *Prescott & Vesterdal, 2013*). The fact that higher values of soil MBC, invertase activity, urease activity, and contents of soil TC, TN, and $NH_4^+$-N were observed in the arrow bamboo root zone than in the spruce root zone (Fig. 1) demonstrated the effects of plant species on soil physicochemical traits. These results were similar to those of a previous study, in which greater activities of invertase and urease and greater contents of TN and microbial carbon were recorded in soils supplied with manure and mineral fertilizers (*Mikanová et al., 2015*). Nitrogen, as the most important nutrient, commonly constrains plant growth due to its low supply in forests, and variations in nitrogen content in soil might affect the ecological succession of forests (*Högberg et al., 2017*). It has been reported that the interaction of nitrogen-fixing microbes with roots could help the plant achieve an efficient $NH_4^+$-N supply in the rhizosphere of spruce-fir forests (*Zhu, Liu & Yin, 2021*). In the present study, the higher $NH_4^+$-N and TN concentrations in the bamboo root zone soil than in the spruce root zone were consistent with previous results reviewed by *Fuke et al. (2021)*. This finding indicates that bamboo roots might stimulate $NH_4^+$-N production and immobilization and benefit the subsequent colonization of spruce trees. Urease catalyzes the hydrolysis of urea to ammonia and is an important soil enzyme that mediates the conversion of organic nitrogen to inorganic nitrogen. Invertase is a ubiquitous enzyme in soils that catalyzes the hydrolysis of sucrose to glucose (*Kandeler et al., 1999*). Urease and invertase are essential soil enzymes for transforming organic carbon and nitrogen into simple carbon and nitrogen sources (*Li et al., 2018*). In this study, the higher activity of urease and invertase in arrow bamboo might lead to greater efficacy of soil carbon and nitrogen transformation, which is also consistent with the higher microbial biomass in the root zone of arrow bamboo (*Diao et al., 2020*). Arrow bamboo regeneration in spruce forests may improve soil properties through the rhizosphere effect, limiting or decreasing soil degradation in spruce plantations.

It is well known that the soil microbial community structure can be altered by plants *via* direct and indirect pathways through root exudates, litter, and nutrient requirements (*Zhou et al., 2017*). In the present study, soil microbial community compositions and functional profiles in the bamboo and spruce root zones were very similar, with differences only in a few phyla, genera, KO, and CAZy gene families. Therefore, the highly similar microbial community and function in the root zones of arrow bamboo and spruce are the result of selection by plant and environmental factors during forest succession.

### Linkage between soil properties and microbial community

Soil properties are determinants of microbial community establishment, which could explain the presence or absence of determined taxa in the rhizosphere, even at very small distances (*Lladó, López-Mondéjar & Baldrian, 2018*; *Zhong et al., 2018*). Soil properties were also tightly associated with microbial functional genes (Table 2, Fig. S4) by studies of secondary successional chronosequences on the Loess Plateau of China (*Zhong et al., 2018*). Our study showed that $NH_4^+$-N concentration was a more important environmental factor than $NO_3^-$-N concentration, significantly influencing microbial functions and microbial taxa (Fig. 7 and Fig. S4 and Table S8). The greater $NH_4^+$-N in the root zone of arrow bamboo evidenced the effects of bamboo roots on $NH_4^+$-N production and immobilization, which in turn affected soil microbes (*Zhu, Liu & Yin, 2021*). As shown by the RDA results (Fig. S4), soil pH, TN, and invertase were the main contributors to the microbial taxa and functions. Soil pH can affect soil nutrient solubility and availability to plants and the soil microbial community (*Naz et al., 2022*; *Zheng et al., 2023*). As an essential nutrient, soil nitrogen is generally the limiting factor for plants in terrestrial ecosystems, resulting in strong competition between microorganisms and plants (*Geisseler et al., 2010*). Nitrogen enrichment shifts the functional genes of microbes related to nitrogen and carbon acquisition (*Treseder et al., 2018*). The trade-off between soil carbon and nitrogen nutrients may have a greater impact on genes in CAZy families (*Cardenas et al., 2018*).

## CONCLUSIONS

The filtration of arrow bamboo roots significantly decreased the diversity of the endophytic microbiome. It made the endomicrobiome structure different from that in its rhizosphere and root zone, with a reduced abundance of bacterial phyla Proteobacteria, Actinobacteria, and Acidobacteria and an increased abundance of fungal phyla Ascomycota, Basidiomycota, and Mucoromycota. *Pseudomonas* was the most abundant and enriched bacterium in the root endosphere. Genes involved in nitrogen fixation, cellulose degradation, and multiple sugar transporters in the root endosphere differ from those in the rhizosphere and root zones of bamboo. The soil TC, TN, $NH_4^+$-N, MBC, and invertase and urease activities in the bamboo root zone were greater than those in the spruce root zone, while the soil microbial community and functional profiles were similar. Soil properties were not related to soil microbial communities but were significantly correlated with microbial functions. The insights into the microbial community and functional structuring gained in this study provide a basis for understanding that the host plant and root zone soil properties select

the root endosphere and rhizosphere/root zone microbiomes of arrow bamboo. Future studies should emphasize root exudate–mediated interactions among microbes in soils affected by arrow bamboo in subalpine coniferous forests.

## ACKNOWLEDGEMENTS

The authors wish to thank Zhong Ping Tang for his assistance in the field and laboratory. We thank the valuable comments of the reviewers, including Dr. José Luis Aguirre-Noyola and Dr. Valeria Souza. We also thank LetPub for its linguistic assistance during the preparation of this manuscript.

### Funding

This study was financially supported by the National Natural Science Foundation of China (32171644), the Sichuan Science and Technology Program (No. 2021YFN0116, 2021YFS0283) and the Maoxian Mountain Ecosystem Research Station of the Chinese Academy of Sciences. ETW was supported by the Sabbatical Project of Instituto Politécnico Nacional, México. The funders had no role in study design, data collection and analysis, decision to publish, or preparation of the manuscript.

### Competing Interests

The authors declare there are no competing interests.

### Author Contributions

- Nan Nan Zhang conceived and designed the experiments, authored or reviewed drafts of the article, and approved the final draft.
- Xiao Xia Chen performed the experiments, analyzed the data, prepared figures and/or tables, and approved the final draft.
- Jin Liang analyzed the data, prepared figures and/or tables, and approved the final draft.
- Chunzhang Zhao conceived and designed the experiments, authored or reviewed drafts of the article, and approved the final draft.
- Jun Xiang performed the experiments, analyzed the data, prepared figures and/or tables, and approved the final draft.
- Lin Luo performed the experiments, prepared figures and/or tables, and approved the final draft.
- En Tao Wang conceived and designed the experiments, authored or reviewed drafts of the article, and approved the final draft.
- Fusun Shi conceived and designed the experiments, authored or reviewed drafts of the article, and approved the final draft.

### Field Study Permissions

The following information was supplied relating to field study approvals (i.e., approving body and any reference numbers):

The experiment was conducted at Maoxian Mountain Ecosystem Research Station of Chinese Academy of Sciences (2019016).

## Data Availability

All the acquired raw sequences are available at the Sequence Read Archive in the Chinese National Microbiology Data Center: NMDC40026007–NMDC40026042.

## Supplemental Information

Supplemental information for this article can be found online at http://dx.doi.org/10.7717/peerj.16488#supplemental-information.

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

# PeerJ

**Food and Agriculture Organization of the United Nations (FAO). 2006.** World reference base for soil resources. World soil resources reports. Rome: FAO. *Available at* https://www.fao.org/soils-portal/data-hub/soil-classification/world-reference-base/en/#:~:text=The%20World%20Reference%20Base%20(WRB,by%20the%20IUSS%20Working%20Group.

**Fuke P, Kumar M, Sawarkar AD, Pandey A, Singh L. 2021.** Role of microbial diversity to influence the growth and environmental remediation capacity of bamboo: a review. *Industrial Crops and Products* **167**:113567 DOI 10.1016/j.indcrop.2021.113567.

**Geisseler D, Horwath WR, Joergensen RG, Ludwig B. 2010.** Pathways of nitrogen utilization by soil microorganisms–a review. *Soil Biology and Biochemistry* **42(12)**:2058–2067 DOI 10.1016/j.soilbio.2010.08.021.

**Gottel NR, Castro HF, Kerley M, Yang Z, Pelletier DA, Podar M, Karpinets T, Uberbacher E, Tuskan GA, Vilgalys R. 2011.** Distinct microbial communities within the endosphere and rhizosphere of *Populus deltoides* roots across contrasting soil types. *Applied and Environmental Microbiology* **77**:5934–5944 DOI 10.1128/AEM.05255-11.

**Guan S, Zhang D, Zhang Z. 1986.** *Soil enzyme and its research methods.* Beijing: China Agriculture Press, 274–297.

**Guzmán-Guzmán P, Santoyo G. 2022.** Action mechanisms, biodiversity, and omics approaches in biocontrol and plant growth-promoting *Pseudomonas*: an updated review. *Biocontrol Science and Technology* **32**:527–550 DOI 10.1080/09583157.2022.2066630.

**Högberg P, Näsholm T, Franklin O, Högberg MN. 2017.** Tamm review: on the nature of the nitrogen limitation to plant growth in Fennoscandian boreal forests. *Forest Ecology and Management* **403**:161–185 DOI 10.1016/j.foreco.2017.04.045.

**Hong CE, Jo SH, Moon JY, Lee J-S, Kwon S-Y, Park JM. 2015.** Isolation of novel leaf-inhabiting endophytic bacteria in *Arabidopsis thaliana* and their antagonistic effects on phytophathogens. *Plant Biotechnology Reports* **9**:451–458 DOI 10.1007/s11816-015-0372-5.

**Hu B, Yang B, Pang X, Bao W, Tian G. 2016.** Responses of soil phosphorus fractions to gap size in a reforested spruce forest. *Geoderma* **279**:61–69 DOI 10.1016/j.geoderma.2016.05.023.

**Hu L, Robert CA, Cadot S, Zhang X, Ye M, Li B, Manzo D, Chervet N, Steinger T, Van Der Heijden MG. 2018.** Root exudate metabolites drive plant-soil feedbacks on growth and defense by shaping the rhizosphere microbiota. *Nature Communications* **9**:1–13 DOI 10.1038/s41467-017-02088-w.

**Jasim B, Joseph AA, John CJ, Mathew J, Radhakrishnan E. 2014.** Isolation and characterization of plant growth promoting endophytic bacteria from the rhizome of Zingiber officinale. *3 Biotech* **4**:197–204.

**Kandeler E, Luxhi J, Tscherko D, Magid J. 1999.** Xylanase, invertase and protease at the soil-litter interface of a loamy sand. *Soil Biology and Biochemistry* **31**:1171–1179 DOI 10.1016/S0038-0717(99)00035-8.

**Kaushal R, Singh I, Thapliyal SD, Gupta AK, Mandal D, Tomar JMS, Kumar A, Alam NM, Kadam D, Singh DV, Mehta H, Dogra P, Ojasvi PR, Reza S,**

**Durai J. 2020.** Rooting behaviour and soil properties in different bamboo species of Western Himalayan Foothills, India. *Scientific Reports* **10**:4966 DOI 10.1038/s41598-020-61418-z.

**Ke PJ, Miki T, Ding TS. 2015.** The soil microbial community predicts the importance of plant traits in plant–soil feedback. *New Phytologist* **206**:329–341 DOI 10.1111/nph.13215.

**Lean EMc, Watson M. 1985.** Soil measurements of plant-available potassium. In: Munson RD, ed. *Potassium in agriculture.* Madison: ASA, CSSA, and SSSA, 277–308.

**Li H, Durbin R. 2009.** Fast and accurate short read alignment with Burrows-Wheeler Transform. *Bioinformatics* **25**:1754–1760 DOI 10.1093/bioinformatics/btp324.

**Li J, Tong X, Awasthi MK, Wu F, Ha S, Ma J, Sun X, He C. 2018.** Dynamics of soil microbial biomass and enzyme activities along a chronosequence of desertified land revegetation. *Ecological Engineering* **111**:22–30 DOI 10.1016/j.ecoleng.2017.11.006.

**Liu C, Wang Y, Pan K, Zhu T, Li W, Zhang L. 2014.** Carbon and nitrogen metabolism in leaves and roots of dwarf bamboo (*Fargesia denudata* Yi) subjected to drought for two consecutive years during sprouting period. *Journal of Plant Growth Regulation* **33**:243–255 DOI 10.1007/s00344-013-9367-z.

**Liu F, Yuan Z, Zhang X, Zhang G, Xie B. 2017.** Characteristics and diversity of endophytic bacteria in moso bamboo (*Phyllostachys edulis*) based on 16S rDNA sequencing. *Archives of Microbiology* **199**:1259–1266 DOI 10.1007/s00203-017-1397-7.

**Lladó S, López-Mondéjar R, Baldrian P. 2018.** Drivers of microbial community structure in forest soils. *Applied Microbiology and Biotechnology* **102**:4331–4338 DOI 10.1007/s00253-018-8950-4.

**Lu MG, Jiang J, Liu L, Ma AP, Leung FC. 2015.** Complete genome Ssquence of Klebsiella variicola strain HKUOPLA, a cellulose-degrading bacterium isolated from giant panda feces. *Genome Announcements* **3**:01200–01215.

**Luo L, Guo M, Wang E, Yin C, Wang Y, He H, Zhao C. 2022.** Effects of mycorrhiza and hyphae on the response of soil microbial community to warming in eastern Tibetan plateau. *Science of The Total Environment* **837**:155498 DOI 10.1016/j.scitotenv.2022.155498.

**Ma JM, Liu SR, Shi ZM, Zhang YD, Chen BY. 2007a.** Quantitative analysis of different restoration stages during natural succession processes of subalpine dark brown coniferous forests in western Sichuan, China. *Ying yong sheng tai xue bao = The Journal of Applied Ecology* **18**:1695–1701.

**Ma M, Jiang H, Luo C, Liu Y. 2007b.** Preliminary study of carbon density, net production and carbon stock in natural spruce forests of northwest subalpine Sichuan, China. *Journal of Plant Ecology* **31**:305–312.

**Martin M. 2011.** Cutadapt removes adapter sequences from high-throughput sequencing reads. *EMBnet. Journal* **17**:10–12.

**Martínez-Romero E, Aguirre-Noyola JL, Bustamante-Brito R, González-Román P, Hernández-Oaxaca D, Higareda-Alvear V, Montes-Carreto LM, Martínez-Romero JC, Rosenblueth M, Servín-Garcidueñas LE. 2021.** We and herbivores eat endophytes. *Microbial Biotechnology* **14**:1282–1299 DOI 10.1111/1751-7915.13688.

**Maulidia V, Sriwati R. 2020.** Endophytic bacteria (genus: *Pseudomonas* spp.) isolated from Aceh bamboo root as biological agent against nematode *Meloidogyne* spp. In: *IOP Conference Series: Earth and Environmental Science*. Bristol: IOP Publishing.

**Mikanová O, Šimon T, Kopecký J, Ságová-Marečková M. 2015.** Soil biological characteristics and microbial community structure in a field experiment. *Open Life Sciences* **10(1)**:249–259 DOI 10.1515/biol-2015-0026.

**Naz M, Dai Z, Hussain S, Tariq M, Danish S Khan, IU, Qi S, Du D. 2022.** The soil pH and heavy metals revealed their impact on soil microbial community. *Journal of Environmental Management* **321**:115770 DOI 10.1016/j.jenvman.2022.115770.

**Olsen SR, Cole CV. 1954.** Estimation of available P in soils by extraction with sodium bicarbonate. USDA Circular. Washington, D.C.: United States Department of Agriculture.

**Peng Y, Leung HC, Yiu S-M, Chin FY. 2012.** IDBA-UD: a de novo assembler for single-cell and metagenomic sequencing data with highly uneven depth. *Bioinformatics* **28**:1420–1428 DOI 10.1093/bioinformatics/bts174.

**Prescott CE, Vesterdal L. 2013.** Tree species effects on soils in temperate and boreal forests: emerging themes and research needs. *Forest Ecology and Management* **309**:1–3.

**Proença DN, Whitman WB, Varghese N, Shapiro N, Woyke T, Kyrpides NC, Morais PV. 2018.** Arboriscoccus pini gen. nov. sp. nov. an endophyte from a pine tree of the class *Alphaproteobacteria*, emended description of *Geminicoccus roseus*, and proposal of *Geminicoccaceae* fam. nov. *Systematic and Applied Microbiology* **41**:94–100 DOI 10.1016/j.syapm.2017.11.006.

**R Core Team. 2020.** R: A language and environment for statistical computing. Version 4.0.2. Vienna: R Foundation for Statistical Computing. *Available at https://www.r-project.org*.

**Saitoh T, Seiwa K, Nishiwaki A. 2002.** Importance of physiological integration of dwarf bamboo to persistence in forest understorey: a field experiment. *Journal of Ecology* **90**:78–85 DOI 10.1046/j.0022-0477.2001.00631.x.

**Shanmuganandam S, Hu Y, Strive T, Schwessinger B, Hall RN. 2019.** Uncovering the microbiome of invasive sympatric European brown hares and European rabbits in Australia. *BioRxiv* DOI 10.1101/832477.

**Sharma A, Singh P, Sarmah BK, Nandi SP. 2020.** Isolation of cellulose degrading endophyte from *Capsicum chinense* and determination of its cellulolytic potential. *Biointerface Research in Applied Chemistry* **10**:6964–6973 DOI 10.33263/BRIAC106.69646973.

**Singh L, Ruprela N, Dafale N, Thul ST. 2021.** Variation in endophytic bacterial communities associated with the rhizomes of tropical Bamboos. *Journal of Sustainable Forestry* **40**:111–123 DOI 10.1080/10549811.2020.1745655.

**Takahashi K, Uemura S, Suzuki J-I, Hara T. 2003.** Effects of understory dwarf bamboo on soil water and the growth of overstory trees in a dense secondary *Betula ermanii* forest, northern Japan. *Ecological Research* **18**:767–774 DOI 10.1111/j.1440-1703.2003.00594.x.

**Treseder KK, Berlemont R, Allison SD, Martiny AC. 2018.** Nitrogen enrichment shifts functional genes related to nitrogen and carbon acquisition in the fungal community. *Soil Biology and Biochemistry* **123**:87–96 DOI 10.1016/j.soilbio.2018.05.014.

**Umemura M, Takenaka C. 2015.** Changes in chemical characteristics of surface soils in hinoki cypress (*Chamaecyparis obtusa*) forests induced by the invasion of exotic Moso bamboo (*Phyllostachys pubescens*) in central Japan. *Plant Species Biology* **30**:72–79 DOI 10.1111/1442-1984.12038.

**Vance ED, Brookes PC, Jenkinson DS. 1987.** An extraction method for measuring soil microbial biomass C. *Soil Biology and Biochemistry* **19**:703–707 DOI 10.1016/0038-0717(87)90052-6.

**Van der Heijden MGA, Martin FM, Selosse MA, Sanders IR. 2015.** Mycorrhizal ecology and evolution: the past, the present, and the future. *New Phytologist* **205(4)**:1406–1423 DOI 10.1111/nph.13288.

**Wang J, Wu Y, Zhou J, Bing H, Sun H. 2016.** Carbon demand drives microbial mineralization of organic phosphorus during the early stage of soil development. *Biology and Fertility of Soils* **52**:825–839 DOI 10.1007/s00374-016-1123-7.

**Wang YJ, Shi XP, Tao JP. 2012.** Effects of different bamboo densities on understory species diversity and trees regeneration in an *Abies faxoniana* forest, Southwest China. *Scientific Research and Essays* **7**:660–668.

**Wang Y, Tao J, Li Y, Yu X, Xi Y. 2007.** Effects of *Fargesia nitida* on species diversity and trees regeneration in different forest cycles of subalpine forest in Wolong Nature Reserve. *Scientia Silvae Sinicae* **43**:1–7.

**Wang YJ, Tao JP, Zhong ZC. 2009.** Factors influencing the distribution and growth of dwarf bamboo, Fargesia nitida, in a subalpine forest in Wolong Nature Reserve, southwest China. *Ecological Research* **24**:1013–1021 DOI 10.1007/s11284-008-0573-2.

**Wei H, Guixiang P, Zhijun X, Shixian C, Zhiyuan T. 2010.** Diversity of endophytic diazotrophs isolated from *Bambusa blumeana* in Guangdong Province. *Chinese Journal of Agricultural Biotechnology* **4**:105–109.

**Woźniak M, Grządziel J, Gałązka A, Frąc M. 2019.** Metagenomic analysis of bacterial and fungal community composition associated with *Paulownia elongata* × *Paulownia fortunei*. *BioResources* **14**:8511–8529 DOI 10.15376/biores.14.4.8511-8529.

**Xiao X, Chen W, Zong L, Yang J, Jiao S, Lin Y, Wang E, Wei G. 2017.** Two cultivated legume plants reveal the enrichment process of the microbiome in the rhizocompartments. *Molecular Ecology* **26**:1641–1651 DOI 10.1111/mec.14027.

**Xu B, Wang J-N, Shi F-S, Wu N. 2016.** Relationships between plant colonization and soil characteristics in the natural recovery of an earthquake-triggered debris flow gully in the Wanglang National Nature Reserve, China. *Journal of Mountain Science* **13**:59–68 DOI 10.1007/s11629-014-3385-6.

**Yan L, Zhu J, Zhao X, Shi J, Jiang C, Shao D. 2019.** Beneficial effects of endophytic fungi colonization on plants. *Applied Microbiology and Biotechnology* **103**:3327–3340.

**Yang C-H, Crowley DE, Borneman J, Keen NT. 2001.** Microbial phyllosphere populations are more complex than previously realized. *Proceedings of the National Academy of Sciences United States of America* **98**:3889–3894 DOI 10.1073/pnas.051633898.

Yao R, Dai Q, Wu T, Yang Z, Chen H, Liu G, Zhu Y, Qi D, Yang X, Luo W. 2021. Fly-over phylogeny across invertebrate to vertebrate: the giant panda and insects share a highly similar gut microbiota. *Computational and Structural Biotechnology Journal* **19**:4676–4683 DOI 10.1016/j.csbj.2021.08.025.

Yuan ZS, Liu F, Liu ZY, Huang QL, Zhang GF, Pan H. 2021. Structural variability and differentiation of niches in the rhizosphere and endosphere bacterial microbiome of moso bamboo (*Phyllostachys edulis*). *Scientific Reports* **11**:1 DOI 10.1038/s41598-020-79139-8.

Zhang NN, Xiang J, Luo L, Rojas Arellano D, Wang YJ, Zhao CZ, Shi FS, Wang ET. 2023. Quantification and diversity of cultivated bacteria in root endosphere and rhizosphere of bamboo species *Fargesia nitida* in association with the tree succession. *Microbiology Research Journal International* **33**:1–16.

Zhang Y, Liu S, Gu F. 2011. The impact of forest vegetation change on water yield in the subalpine region of southwestern China. *Acta Ecologica Sinica* **31**:7601–7608.

Zheng X, Wu Y, Xu A, Lin C, Wang H, Yu J, Ding H, Zhang Y. 2023. Response of soil microbial communities and functions to long-term tea (*Camellia sinensis* L.) planting in a subtropical region. *Forests* **14**:1288 DOI 10.3390/f14071288.

Zhong Y, Yan W, Wang R, Wang W, Shangguan Z. 2018. Decreased occurrence of carbon cycle functions in microbial communities along with long-term secondary succession. *Soil Biology and Biochemistry* **123**:207–217 DOI 10.1016/j.soilbio.2018.05.017.

Zhou Y, Zhu H, Fu S, Yao Q. 2017. Metagenomic evidence of stronger effect of stylo (legume) than bahiagrass (grass) on taxonomic and functional profiles of the soil microbial community. *Scientific Reports* **7**:1–11 DOI 10.1038/s41598-016-0028-x.

Zhou Z, Yu M, Ding G, Gao G, He Y. 2020. Diversity and structural differences of bacterial microbial communities in rhizocompartments of desert leguminous plants. *PLOS ONE* **15**:e0241057 DOI 10.1371/journal.pone.0241057.

Zhu W, Lomsadze A, Borodovsky M. 2010. Ab initio gene identification in metagenomic sequences. *Nucleic Acids Research* **38**:e132–e132 DOI 10.1093/nar/gkq275.

Zhu X, Liu D, Yin H. 2021. Roots regulate microbial N processes to achieve an efficient $NH_4^+$ supply in the rhizosphere of alpine coniferous forests. *Biogeochemistry* **155**:39–57 DOI 10.1007/s10533-021-00811-w.

Zhu L, Yang Z, Yao R, Xu L, Chen H, Gu X, Wu T, Yang X. 2018. Potential mechanism of detoxification of cyanide compounds by gut microbiomes of bamboo-eating pandas. *MSphere* **3**(3):e00229-18 DOI 10.1128/mSphere.00229-18.