# Peer review of "Rhizocompartmental microbiomes of arrow bamboo (Fargesia nitida) and their relation to soil properties in Subalpine Coniferous Forests"

_PeerJ, doi:10.7717/peerj.16488_

## Round 0.1 · original submission · Major Revisions

Your manuscript was evaluated by three experts that made very good reviews and agree that, while original an interesting, the manuscript requires a major revision before it can be accepted.

They sent different comments for each section of the manuscript, recommend changes and improvements, and asked you to review and cite additional and pertinent literature.

Please, consider all their comments, make the pertinent changes in a new version of the manuscript and answer them point by point in a detailed letter.

Reviewer 1 ·

Basic reporting

The aim of the present paper was to investigate the diversity and function of the microbiome in the root zone associate with arrow bamboo and spruce in China. This subject could be according to the scope of Peer J. However, the actual version has several problems. The authors must improve the introductions adding the conceptual bases of previous studies, to establish hypotheses. Additionally, the description of the experimental designed must be strongly improved and the author must explain all statistical analyses used. Finally, the narrative history of discussion must be also improved. Therefore, the paper could be accepted after major revision.

In detail:
1. Abstract:
Minor comments.
L33: add the dominant fungal phyla.
L41-42: It is not clear what is meaning the effect of NH4, Is it a correlation? Please clarify.
L45-47. These sentences are not clear.
L47-48: Improve this sentence as a conclusion of the study.

2. Keywords: avoid the wors included in the title: Arrow bamboo, Subalpine
Forest

3. Introduction
The author must add the previous studies of processes that explain the biodiversity in the root zone, as a two-step model of microbiome selection proposed by Bulgarelli et al. Se references below:
Bulgarelli, D., Garrido-Oter, R., Münch, P. C., Weiman, A., Dröge, J., Pan, Y., et al. (2015). Structure and function of the bacterial root microbiota in wild and domesticated barley. Cell Host Microbe 17, 392–403. doi: 10.1016/j.chom.2015.01.011
Bulgarelli, D., Rott, M., Schlaeppi, K., Ver Loren van Themaat, E., Ahmadinejad, N., Assenza, F., et al. (2012). Revealing structure and assembly cues for rabidopsis root-inhabiting bacterial microbiota. Nature 488, 91–95. doi: 10.1038/nature11336
Bulgarelli, D., Schlaeppi, K., Spaepen, S., van Themaat, E. V. L., and Schulze-Lefert, P. (2013). Structure and functions of the bacterial microbiota of plants. Annu. Rev. Plant Biol. 64, 807–838. doi: 10.1146/annurev-arplant-050312-120106
Barajas, H.R., S. Martínez-Sánchez, M.F. Romero, C. Hernández, L. Servín-González, M. Peimbert, R. Cruz-Ortega, F. García-Oliva and L.D. Alcaraz 2020. Testing the two-step model of plant root microbiome acquisition under multiple plant species and soil sources. Frontiers in Microbiology 11: 542742. doi: 10.3389/fmicb.2020.542742.

4. Materials and Methods.
The description of the experimental designed must be strongly improved. The author must explain all statistical analyses used. I also recommend, that the authors analyze the interaction between the microbial community and soil variable with a Redundance analysis.

L108-110: Add the author to the scientific name of the plant species mentioned.
L111: what is means 'id numbers 2019016. This information is relevant for this study. I recommend moving to financial support section.
L116-127: improve description of sampling design. It is not clear how many replicates were: nine (sampling plots) or 5X9 = 45?
L178-179: Improve the description of One-way ANOVA: which treatments were analyzed? The tree treatment associated with bamboo and spruce samples (4 treatments) or only associated with bamboo. I recommend that the ANOVA was done only the treatments associated with bamboo, and a t-Student test for comparing root soil of bamboo and spruce.
L182-186: I recommend a Redundance analysis for correlation between microbial community composition and soil variables, rather than Spearmen Correlation.

5. Results.
Minor comments below:
L189-191: Were these comparisons done with One-Way Anova? the authors must use t-Student test (see my comments in method section of statistical model used).
L191-193: I recommend if the author did not find statistical differences, they avoid describing the results.
L258: The Mantel test was not mentioned in the method section. I recommend using Redundance analyses.


6. Discussion.
The narrative history of discussion must be also improved. The actual version of discussion is too long and speculative. I recommend that the conceptual bases of previous studies be clearly presented in the introduction section, in order to establish hypotheses. This will make it easier for the authors to interpret the results in the discussion section.

Experimental design

The description of the experimental designed must be strongly improved. The author must explain all statistical analyses used. I also recommend, that the authors analyze the interaction between the microbial community and soil variable with a Redundance analysis.

Validity of the findings

The actual version of discussion is too long and speculative. I recommend that the conceptual bases of previous studies be clearly presented in the introduction section, in order to establish hypotheses. This will make it easier for the authors to interpret the results in the discussion section.

Additional comments

no comments.

·

Basic reporting

The authors present a manuscript on the diversity and functionality of the microbiome across different root compartments of Fargesia nitida, as well as the comparison of the root zone of this bamboo with that of Picea asperata. Enzyme activities and soil characteristics were determined and correlated with metagenomic data. The authors describe the changes between compartments with emphasis on genes encoding enzymes for C and N cycling and CAZymes. The manuscript is well written, and there is an appropriate development of the research.

Experimental design

The experimental design is correct. The analyses have replicates and the sequencing data were deposited in a public repository. However, details about the bioinformatics analysis are not explained. Comments about this aspect are taken up in the comments section of this review.

Validity of the findings

The results shown are relevant and statistically well-supported. However, the analyses at the phylum level do not allow to recognize the microbial groups that are changing between compartments, and it was not determined to which bacterial or fungal genus the genes encoding the differentially abundant enzymes belong.

Additional comments

Here are some comments that I believe will help to improve the manuscript.

ABSTRACT

L-28 Properly name rhizocompartments to avoid confusion. For example, put "root endosphere" instead of "root endophytes".

L-44 Check spelling, e.g., rhizopsphere / rhizosphere.

Be much more specific about how the microbial communities are different, diversity, functions, etc., and the trends you observed across compartments.

Mention the outstanding differences between the root zone of Fargesia nitida and Picea asperata.

INTRODUCTION

The introduction is appropriate to the manuscript topic. I recommend that you include recent references to the mechanisms that plants use to govern the assembly of communities in the rhizosphere, especially the effect of root exudates.

MATERIALS AND METHODS.

L-147 Include the units in which cellulase activity is reported.

L-148 Please specify the amount of sample used for DNA extraction from each rhizocompartment. In the case of the endosphere, only a small portion of the root was used?

Your methodology needs more details in bioinformatics analysis. Please address the following comments.
-What were the criteria for defining good quality reads?
-Which genomes were used to map the host sequences?
-What were the parameters used to assemble with IDBA-UD?
-It is not explained how the taxonomic classification of the microbiota was performed, nor which were the taxonomic reference programs and databases.
-Details about the tools used in QIIME, and the version used?
-Please mention how the alpha diversity indices were determined and how they were compared between treatments. They appear in results but were not calculated.
-I suggest that your beta diversity analyses were supported by PERMANOVA tests.

One of the limitations of the study is the missing information on the bulk soil microbiota of the study area. By not having this information, the first filter of selection by root exudates on the diversity of the native soil microbiota is unknown. Why was the bulk soil microbiota not analyzed?

When I checked the data repository at NMDC they indicate that the metagenomes were obtained by sequencing with illumina Novaseq6000, but this does not match the methodology described here, which says it was by HiSeq X-ten platform. Why does the information not match?



RESULTS

L-199 The endosphere shows an enrichment of eukaryotes compared to the other compartments. In this case, are these really eukaryotic microorganisms or is it host plant DNA that was sequenced? Please specify for endosphere what percentage of the raw reads correspond to plant and how many to microorganisms.

Please give the justification for determining urease and invertase activity and no other enzymes to evaluate soil microbial metabolism. In L-356 you mention that they are important, but do not explain the processes they are catalyzing?

L219 It is necessary to clarify the domain to which each microbial genus belongs. The recommendation is to show the results distinguishing between bacteria, eukarya and archaea, and to avoid mixing them.

L-226, I suggest that the alpha and beta diversity analysis be included as a figure in the results.

L230 In section 3.3. “Functional gene profiles of the microbial communities" shows that the abundance of diverse enzymes varies between samples, however, it is not mentioned to which microbial genera these activities are assigned. Knowing this is relevant to determine if changes in diversity are linked to changes in functionality in each rhizocompartment or if there are functional redundancies and among which microorganisms. Please include the taxonomic assignment of the most differentially abundant enzymes.

L257 In section 3.4. "Relationship between soil properties and the microbiomes", you should consider showing the correlations with more statistical support and detailing them in the manuscript, in order to identify possible key microbial groups in the enzyme activities evaluated.

In the main results, a graph or table showing differences at the genus or family level is needed. Most of the differences were explained at the phylum level and this does not allow us to know which microbial groups were filtered between compartments.

The comparison between root zone of Fargesia nitida and Picea asperata was little explored. I suggest that you indicate which taxonomic groups are exclusive in each plant in order to know if there is an effect of host genotype on soil microbiota.

DISCUSSION

L-287. It was mentioned that the increased abundance of fungi in the roots is due to mycorrhizae. Could you demonstrate that the genera you mention are the most abundant in the ndosphere vs. rhizosphere and root zone?

L298 prdominant/ predominant

L309 The role of endophytes as colonizers of hervibore microbiota has recently been reviewed. doi: https://doi.org/10.1111/1751-7915.13688

It is often mentioned that changes in the abundance of genes encoding enzymes is a consequence of the distribution of specific microbial groups, e.g., L327 "The significantly low abundances of cellulose and hemi-cellulose genes, but greater chitinase genes in root endosphere compared with that in rhizosphere and root zone soils of arrow bamboo may be related to the distribution of fungi in these three rhizocompartments".
However, there is no mention in the text or the tables to which microbial group these genes are associated. If you have the sequences that code for the enzymes, you could determine this, that would help you to have much more solid results about who does what, and avoid speculation.

Discuss the reason why there is correlation between soil enzyme activities with the genes detected, but not with the structure of the microbial communities.

CONCLUSIONS

Be much more specific about how the microbial communities are different, diversity, functions, etc., and the trends you observed across the compartments.

·

Basic reporting

The text has many grammatical errors as well as errors in translation, it needs to improve English and the flow of the text. Some parts are written badly and the reader gets confused, such as the part of the diazotrophs. It is not clear what the authors ment.

Experimental design

Experimental design is correct and proper replicates have been done. Nutrients as well as microbial diversity were assess, inside the root (BE), in the rhizosphere (BR) soil around the root (BZ) and spruce related soil (SZ). Also enzymes were tested. 36 metagenomes. Samples showed different diversity as expected as well as different functions. The results are interesting and valid, but more clarity is needed in the text.

Validity of the findings

The findings are valid and the study is well done.

Additional comments

I think if the text English is improved and delete most of the acronyms (BE, BR, Bz, Sz) and make the text flow as to explain the story it will be a great improvement.
The term filtration is used out of context and probably due to a bad translation, the correct term is environmental filtering and is not a "gate" is caused by selection from within the plant root system.

---

## Round 0.2 · Minor Revisions

The three previous reviewers agree that you made an important effort, and they are happy with the new version of your paper.

Reviewer 2 has some minor addition comments.
Please consider them:
1) Move table 2 "PERMANOVA analysis" to supplementary material.
2) L-169 QIIME 19.1: Do you mean “1.9.1”?
3) Make sure that the final figures are of high quality for the final version.

I have some additional minor comments that I would like you to evaluate and make.

Abstract:
Line 39 (and the rest of the paper): “Blank soil”: It is not really a “blank”, but is the soil of the spruce root area… Change to something like: “adjacent soil”.

Introduction:
Line 72: “main synusia”: Even if I consider myself an experienced plant ecologist, I do not know what this means. If you like this word, explain it: “main synusia (i.e., the main plant communities)”

Line 76: change “act as” to “is”, and change “for” to “of”, so it reads as “Fargesia bamboo is the main food source of the endangered…”

Line 80: Change “Arrow bamboo is usually recovered as a pioneer” to “Arrow bamboo is a pioneer…” .

Line 86: Give the Latin name of the “moso bamboo” and indicate where it is found or where it was studied.

Line 89: Change “in subalpine coniferous forests remain unclear” to “in subalpine coniferous forests has not been described (or studied)”.

Materials and Methods:

Line 111: Change to “which also serve as blank control under the forest.” to “which also serves as a control of the microbiome of the soil”.

Line 113: I do not understand what you mean by: “were demarcated in the transit area of the alpine forest (mixed-coniferous forest).” You may change it (if it is OK) to: “were set-up (or implemented, or sampled) in a transition area between mixed and coniferous forest”.

Results:
Lines 204-205: I do not understand this section: “Archaea accounted for 0.07%, presenting significantly in the order of root endosphere < rhizosphere = root zone of…”. Perhaps you mean something like: “Archaea accounted for 0.07% of the root endosphere (or of the total analysis, it is not clear!), being significant lower in percentage in the root endosphere than in the rhizosphere, which was not different than the root of the bamboo”

Discussion:

Line 308: Give the Latin name of Aceh bamboo, and indicate where it is found or where it was studied.

Line 310: Give the Latin name of Guangdong bamboo, and indicate where it is found or where it was studied.

Line 315: Change “The dominant genera in” to “The dominant bacteria genera in…”
Line 370: Something is missing in “spruce result selection by”, perhaps you mean “… root zones of arrow bamboo and spruce are the result of selection by plant and by environmental factors…”

Acknowledgments:

You may want to acknowledge the efforts of the three reviewers, two of them are giving their names.

=======

Reviewer 1 ·

Basic reporting

The aim of the present paper was to investigate the diversity and function of the microbiome in the root zone associate with arrow bamboo and spruce in China. This subject could be according to the scope of Peer J. The authors adequately responded to the reviewers' comments and substantially improved the Ms. Therefore, the paper could be accepted.

Experimental design

no comments.

Validity of the findings

no comments.

Additional comments

no comments.

·

Basic reporting

The quality of the paper improved and shows relevant results for the understanding of the microbiota assembly and functionality in different compartments associated with plant roots.

Experimental design

The experimental design is appropriate for the scope of this work.

Validity of the findings

The statistical analyses now shown give much stronger validity to the results.
A suggestion is that table 2 "PERMANOVA analysis" be moved to supplementary material.

Additional comments

L-169 QIIME 19.1, did you mean is 1.9.1?
Make sure that the final figures are of high quality for the final version.

·

Basic reporting

The manuscript is interesting and novel

Experimental design

The manuscript of the revised version is very clear, I feel they did all the suggested changes

Validity of the findings

I fell the experimental caveats have been solved

Additional comments

I had accepted the first version with minor changes, this version I fell is ready

---

## Round 0.3 · accepted · Accept

Thank you for the changes and corrections in the manuscript.

I consider that your manuscript is now ready for publication.

I believe it is an important contribution to our understanding or the plant-microbe relationships and of the microbiomes involving an interesting species, and I want to acknowledge your efforts and congratulate all the authors for this relevant study.